# A Dimensional Comparison between Evolutionary Algorithm and Deep Reinforcement Learning Methodologies for Autonomous Surface Vehicles with Water Quality Sensors

**DOI:** 10.3390/s21082862

**Published:** 2021-04-19

**Authors:** Samuel Yanes Luis, Daniel Gutiérrez-Reina, Sergio Toral Marín

**Affiliations:** Department of Electronic Engineering, University of Seville, 41009 Seville, Spain; dgutierrezreina@us.es (D.G.-R.); storal@us.es (S.T.M.)

**Keywords:** Unmanned Surface Vehicles, Evolutionary Algorithm, Deep Reinforcement Learning, machine learning, intelligent sensor system

## Abstract

The monitoring of water resources using Autonomous Surface Vehicles with water-quality sensors has been a recent approach due to the advances in unmanned transportation technology. The Ypacaraí Lake, the biggest water resource in Paraguay, suffers from a major contamination problem because of cyanobacteria blooms. In order to supervise the blooms using these on-board sensor modules, a Non-Homogeneous Patrolling Problem (a NP-hard problem) must be solved in a feasible amount of time. A dimensionality study is addressed to compare the most common methodologies, Evolutionary Algorithm and Deep Reinforcement Learning, in different map scales and fleet sizes with changes in the environmental conditions. The results determined that Deep Q-Learning overcomes the evolutionary method in terms of sample-efficiency by 50–70% in higher resolutions. Furthermore, it reacts better than the Evolutionary Algorithm in high space-state actions. In contrast, the evolutionary approach shows a better efficiency in lower resolutions and needs fewer parameters to synthesize robust solutions. This study reveals that Deep Q-learning approaches exceed in efficiency for the Non-Homogeneous Patrolling Problem but with many hyper-parameters involved in the stability and convergence.

## 1. Introduction

Ypacaraí Lake, as the biggest water resource in Paraguay, holds a high importance for the surrounding cities, their industries and tourism. Because of the economical development of the near populations, the absence of sewerage infrastructure over the years and the growing tourism, the waste dumped into the water has dramatically increased. This aggravated contamination situation caused in the Lake an unnatural process called eutrophication, which consists in an excessive concentration of nutrients (phosphorous and nitrates). The lake ecosystem, with a very rich composition for cyanobateria to grow, has experienced an abnormal blooming of blue-green colonies all around the lake surface (see Figure 1). This kind of bacteria is proven to be harmful for the local fauna and humans, and causes the deoxygenation of the waters, fetid smells, and other undesirable effects [1]. The situation becomes even harder for the authorities that are required to close the lake to tourists because the blooming is both cyclic and unpredictable. The only solution to this problem comes from the hand of a multidisciplinary approach, including public investment in infrastructure, biochemical studies of the waters, etc.

One mandatory problem to solve is having an updated biological status of the waters with an efficient and periodic sampling in specific zones of interest. As Ypacaraí Lake is very large (60 km${}^{2}$), the manual sampling process would be too inefficient and slow to be useful for the biologists, even worse when the cyanobacteria situation changes within the days. Therefore, a reactive monitoring system is required. In a resourceful solution, the use of Autonomous Surface Vehicles (ASVs) with a water-quality electronic sensor modules has been proposed by several authors as an excellent and efficient solution to monitor natural environments [2,3,4]. Equipped with turbidity sensors, pH-meters, dissolved-oxygen modules, etc, an ASV could supersede easily a static sampling station, providing a more efficient way of measuring the contamination. As a matter of fact, the utilization of these automated agents entails a less-costly implementation because of the cheapening of the unmanned robots in recent years. Regarding the autonomy, it is also possible to launch missions long enough because of the small size of the ASVs, which are propelled by electric motors and batteries.

Nevertheless, this solution requires a sophisticated strategy to cover efficiently the different zones of the Lake. In order to guarantee real efficiency, the ASVs must be autonomous and their missions non-supervised. Regarding the mission, every autonomous vehicle is programmed to execute: (i) a local path planning (also known as a trajectory interpolation) and (ii) a global path planning. On the one hand, The local planning consists of tracing the physical route from a specific point to another, avoiding obstacles in the middle of the trajectories [5]. Sensors like LIDARs, cameras, and so on, are used to adapt the path calculated by the global planner to unforeseen situations like the appearance of moving obstacles. On the other hand, the global path planning, which is the problem here discussed, is a high-level task that consists of deciding the optimal locations or waypoints to achieve the objective based on different optimization criteria, like the redundancy of visitation or the maximization of the coverage area. This global planning must not only consider constraints of the terrain like the Ypacaraí non-navigable zones, battery of the ASVs, etc, but also take in account the optimality of the task when adapting the path to possible changes in the scenario. Therefore, the global path planning must consider a more sophisticated strategy than the local planner to decide where to travel. It must complete the task not only with the available scenario information, but also with an inferred model of the environment. Once the global path planner has solved the target waypoint, the local planner is in charge of the movement of the actuators to reach it. Moreover, the multi-agent scenario is a natural extension of the given problem because of the large size of Ypacaraí Lake. The use of multiple vehicles to explore and supervise the environment is mandatory if an exhaustive monitoring is pursued, hence coordination is needed between the agents. With multiple-agents it is not only important to consider the exploration of specific and important zones of the lake but also to avoid collisions between them.

Regarding the monitoring task, it is required to visit those zones of high importance but also to revisit them periodically because the green-blue algae blooms change within time. This periodicity requirement adds complexity to the global planning because the agent must design its paths considering if it is worth revisiting zones or visiting new ones. This problem, named the Patrolling Problem [6] or the Vigilant Problem, is addressed commonly with a metric graph framework. Thus, each worth-visiting zone of the environment area is represented as a node in a discrete map [7]. In the particular problem where the importance of the zones are weighted according to an environmental criterion, the problem is referred as the Non-Homogeneous Patrolling Problem (NHPP) [6].

Considering the morphology of the Lake, the size of the paths, the importance of every worth-visiting zone and the number of agents, the complexity of this task scales up easily to an unfeasible amount of solutions (NP-hard). In regards to the dimensionality of the problem, it is sometimes necessary to use an approach based on heuristics methods to find acceptable routes to synthesize a complete path planning to monitor the lake. This work addresses the global path planning for a fleet composed by 1 to 3 ASVs and using the most common and powerful approaches found in the literature: the meta-heuristic methodology given by Evolutionary Algorithms (EA) [8] and the Artificial Intelligence method of Deep Reinforcement Learning (DRL) [6]. Both approaches tend to optimize the same function, i.e., the episodic reward, which is also known as the fitness function, for the paths in an ASV mission. Within the context of the Patrolling Problem, the reward function will evaluate the paths of the ASVs in terms of the redundancy of visiting, collision avoidance, and other criteria. Both methods are off-model algorithms, so there is no need of any previous dynamic model or any different previous knowledge of the scenario rather than this aforementioned reward function. This work compares the performance and convergence in both cases related to the number of possible states-action combinations: the number of movements the agent (one ASV) can perform, the size of the map, the number of ASVs and, finally, the reactivity to changes in the mission conditions. Additionally, this work could serve as an empirical guide to select the best approach depending on the dimension requirements of the task, the level of performance needed for the patrolling and the level of reactivity required for the ASVs.

The main contributions of this paper are:A (μ+λ) Evolutionary Algorithm and a Deep Q Learning approach for the resolution of the Non-Homogeneous Patrolling problem in Ypacaraí Lake.A performance comparative analysis of the sample-efficiency metric and computation time using non-intensive computation resources.An analysis on the reactivity and generalization of the solutions provided by the tested methodologies for different resolutions, number of agents and changing environments.

This work is organized as follows. Section 2 describes the state of the art and presents relevant works and approaches that address similar monitoring problems. Section 3 is dedicated to the Patrolling Problem and the assumptions taken into account to design a simulator to test the methodologies. Section 4 describes both methods: (μ+λ) Evolutionary Algorithm and Double Deep Reinforcement Learning (DDRL) and the state representation. Then, the results are presented and discussed in Section 5 for different dimensions of the map, changes on the environment and fleet sizes. In Section 6, the conclusions are explained and future lines of work suggested.

## 2. Related Work

The use of unmanned vehicles in environmental monitoring has been a common proposal in recent decades because of the improvements in the manufacturing costs, the increasing computational capabilities of electronic systems, wireless communications and batteries performance [4]. In the particular case of the monitoring task of hydric environments, such as lakes and water reservoirs, the use of Unmanned Surface and Underwater Vehicles brings the opportunity to explore large spaces without human intervention. An implementation example can be found in [9], where an ASV with a double-catamaran morphology is tested to obtain the bathymetric profile of rivers and coastlines. Apart from the motors, these vehicles are usually well-equipped with: water quality sensors to make measurements along the planned path, a localization system, and a communication module to report the results of the mission. In [10], a Hybrid Surface-Underwater Vehicle prototype is designed in order to study the nitrates concentration in Norwegian fjords. This evidences the suitability of such vehicles compared to human manual sampling because of the small size of the prototype, the relatively low-cost and the lower level of environmental intrusion. The same problem is pursued in [11], where it is designed a garbage-collector based on an ASV to clean water resources, and at the same time, measuring some water quality variables like pH.

Regarding the Ypacaraí monitoring task, it is remarkable how the use of ASVs has provided researchers the possibility to analyze such an big water resource efficiently. In [2,3,5,6] are proposed the use of a robotic catamaran-like ASV to survey the environmental metrics of the Ypacaraí waters, addressing different tasks (see Figure 2 for an image of the ASV).

Depending on the scenario and the objectives of the mission, the global path planning algorithm of every ASV must face many challenges and constraints in order to obtain an implementable algorithm. It can be considered, for example, the battery consumption constraint like in [12], where the objective is to maximize the coverage under a budget number of waypoints. Some works like [2] also consider a dynamic scenario where there is a temporal dependency of the achievable optimality. Some environments, like the one considered in [13], address the incomplete observability of the environment where part of the scenario remains unknown. In the latter case, the problem is even harder because of the lack of complete observability, which is a specific condition where the state variables cannot be measured every time. These typical problems, with a high number of possibilities, encourage the use of Artificial Intelligence or heuristic algorithms, commonly represented by two leading approaches: the EAs [14,15,16] and Reinforcement Learning (RL) algorithms [6,12,17].

### 2.1. Deep Reinforcement Learning in Path Planning

Since the results of [18,19] were presented, a common approach has been the use of DRL to solve path planning problems because of the ability of the algorithm to deal with high dimensional inputs and stochastic scenarios. Using the foundations of learn-by-trial and thanks to the ease of Neural Networks to adapt to high dimensions, this particular approach has been proven to be effective in the global path planning for autonomous vehicles. For example, in [12], a Deep Q-Network (DQN) is used to estimate the optimal actions that a quad-copter should take in a global path planning to maximize the area considering also battery constraints. In [20], a remarkable use of DRL algorithms can be found. A combination of the value iteration and policy iteration methods, which is called Actors Critic with Experience Replay, is used to compute the path for a multi-body hexagonal robot to complete the covering of an area. In [21], another novel Deep Q-Learning approach is implemented to successfully design global paths for a multi-agent deploy of a flying ad hoc network (FANET) with drones. Ref. [22] addresses the particular case of an autonomous underwater vehicle, where it is compared a DQN architecture with a Deep Deterministic Policy Gradient (DDPG) approach for continuous action space. Another interesting application of such continuous optimization algorithms can be found in [23,24] where it is used a Soft Actor-Critic architecture to improve the thermal and energetic efficiency of a vehicle under power constraints. For some applications like the one presented in this paper, there is no need of a continuous domain and a discretized space is enough, but it is remarkable how the DRL can adapt to these problems as well. Finally, in [6], the authors proposed the resolution of the patrolling problem in the Ypacaraí Lake scenario. The DRL approach tested in this paper is based on its results and parametrization.

When addressing the multi-agent planning in DRL, there has been recent advances in the cooperative control of autonomous agents for path planning [25,26,27,28,29]. Ref. [26] addresses the competitive behavior of agents to optimize the performance in the predator-prey toy problem using an actors-critic morphology, which is suitable for different agents action spaces. In [27], the cooperation between several agents is pursued with a convolutional DRL network, similar to the proposed in [25]. As in this work, it uses a graph framework for the resolution, as one of the most common formulations of the space to solve with RL. Ref. [28], proposed a multi-agent local path planner to solve the obstacle avoidance and coordination of multiple unmanned aerial vehicles (UAVs). This latter case is an example of a continuous action domain application of a value iteration methodology for multiple agents. The work in [29], used a combination of convolutional networks with a Proximal Policy Optimization (PPO) to solve a the multi-agent planning in an urban environment of autonomous cars. This work also focuses on the effect of some important parameters in the performance of DRL and proves the need of hyper-parametrization as it is intended in this proposal.

### 2.2. Genetic Algorithms in Path Planning

The EA takes advantage of bio-inspired evolutionary foundations to synthesize increasingly optimal paths within every generation. Several authors deal with high dimensional planning problems in vehicles [2,7,30,31,32]. In [30], the particular case of the Ypacaraí area coverage is treated as a Traveling Salesman Problem (TSP). Ref. [2] also proposes a reactive methodology to monitor sudden blossoms of cyanobacteria in the same Lake. Ref. [7] studies the homogeneous Patrolling Problem using both EA and RL methodologies and, as it is proposed in this paper, it uses a graph formulation of the problem. It concludes that the EA and RL methods outperform other methodologies by far. Some other works, like [31], put the focus on a high action space considering also several physical constraints. This particular work addresses a 3D scenario with a quad-copter as the agent, imposing severe safety restrictions to the movement, which highlights the ability of genetic algorithms to robustly synthesize good solutions even when the restrictions are hard. In this sense, [15] proposes the use of such genetic algorithms for planning in robotic manipulators, where the number of degrees of freedom (DOFs) is extremely high. Ref. [32] also addresses the path planning via a Natural Language Generation (NGL) using an Evolutionary methodology for many agents to complete different tasks. This work also addressed the dimensionality by comparing the number of the possible states and the computation cost.

For the multi-agent case, EA has been used in several works to solve coordinated path planning [33,34,35]. For example, [33] proposes an extended genetic approach for path planning of multiple mobile robots with obstacle detection and avoidance in static and dynamic scenarios. For the same task, ref. [34] focuses on coordinated behavior using decentralized independent individuals in a static graph-like map. These approaches apply directly the single-agent genetic algorithm to multiple robots by expanding the chromosome representation of the individual by including all the robots paths. This is the method used in this paper for comparison purposes, since it is a recurrent way for dealing with the multi-agent scenario. Finally, in [35], a generalization approach for multi-agent path planning is proposed using genetic algorithms.

### 2.3. DRL and EA Comparison Overview

In general terms, in order to solve NP-problems, genetic algorithms provide a black-box optimization methodology easy to implement and at a relatively low computational cost. Despite that, the recent development of graphic computation and the new computation architectures (such as Graphic Processor Units or Tensor Processor Units) has caused the RL algorithms to excel in solving these dimensional unfeasible problems. In [36], the reader can find an exhaustive analysis of the advances in both methodologies and other novel hybrid approaches. This work states that EA and RL could serve as off-model optimizers, but RL (and DRL above all) tends to be more reactive and provides a better generalization of the task. On the contrary, most of the DRL methods, because they rely on non-linear estimators such as Neural Networks (NN), have no theoretical guarantee of stability in the learning [37]. The task addressed in [38] discussed the need of exhaustive comparisons between these two approaches, since there is few empirical results available. What is also compared is the performance of Genetic Algorithms (GA), Time Difference methods (Q-learning alike) and one of the most remarkable hybrid methodology: Neuro-Evolution of Augmenting Topologies (NEAT), where the policy network is optimized using EAs.

This paper focuses on the analysis of the performance of both methods related to the dimension of the problem as a current need, since there is none comparative studies at this matter. Where other works like [7] focus on the solution performance only or in the application of hybrid EA-RL techniques like [39], where it is proposed a evolutionary temporal difference learning, this study addresses how the dimension of the action-state domain of a single agent problem affects the convergence and generalization of both DRL and EA. Table 1 and Table 2 show a summary of significant contributions to path planning with DRL and EA in both single-agent and multi-agent cases.

## 3. Statement of the Problem

This section describes the non-homogeneous patrolling problem for Ypacaraí Lake and the undirected graph assumption, among other characteristics of the scenario. Regarding the scenario, Ypacaraí Lake has a surface of approximately 60 km${}^{2}$ and its shores overlook the cities of San Bernardino, Areguá and Ypacaraí. The contamination dumped in those places makes the cyanobacteria bloom in certain zones, causing the deoxigentation and acidification of these areas. Therefore, these are zones of high probability to be contaminated and they must be supervised more thoroughly than others. This entails the definition of an interest map that will define the absolute importance of every zone which means a priority to be visited. Figure 3 depicts a satellite image of Ypacaraí Lake (left) and the importance map used to apply the Patrolling Problem. This specific map could be obtained after localizing the maximum points of interest of a contamination index or by using expertise knowledge of the mandatory points to survey. In this particular work, we used an interest function very similar to the benchmark used in [3] for the Ypacaraí environmental variable estimation.

### 3.1. The Patrolling Problem

In order to apply the patrolling problem to the Ypacaraí scenario, a discrete map of the lake is defined with same size cells, like pixels in an image, with different resolutions (N). Let G(V,E,W) be an undirected graph of nodes *V* (corresponding to the physical cells of the discrete map), edges *E*, and an idle value *W*. If the agent is in a cell, it can perform an action *a* to travel to a surrounding cell once every step (from a total *T* steps) and only if the aforementioned is navigable (is a water cell) (see Figure 4). It is also assumed that the graph is metric and diagonal movements are allowed. The idle value *W* represents the number of time steps a certain node has been unvisited since the last visit. This way, the Patrolling Problem is defined as the task of finding the path that minimized the average idleness of the graph in *T* possible movements
(1)$$P^{*}:=\left\{a_{1},\ldots ,a_{T}\right\}\to \min_{P}\sum_{i=1}^{\left|V\right|}\left(\frac{1}{\left|V\right|}W_{i}\right)$$

This problem suits the monitoring task in Ypacaraí Lake because the agents need to revisit periodically the zones to update its measurements. In fact, the Patrolling Problem can be extended to the non-homogeneous Patrolling Problem (NHPP) by imposing an importance criterion to every cell of the map. Since the visiting periodicity requirement has to consider that there are zones more interesting than others because of the blooms of cyanobacteria or because these have a higher chance of being contaminated, the graph can be pondered with an importance term I∈[0,1]. This term will weight the acquired idleness *W* to obtain a pondered idleness $W_{r}=W\times I$ meaning that higher zones (those with higher I) will be worthier of visiting than other with lower importance. Defining a path P with *T* number of actions (movements), we defined the NHPP as:
(2)$$P^{*}:=\left\{a_{1},\ldots ,a_{T}\right\}\to \min_{P}\sum_{i=1}^{\left|V\right|}\left(\frac{1}{\left|V\right|}W_{i}\times I_{i}\right)$$

### 3.2. Simulator and Assumptions

It has been implemented a scenario simulator that represents the morphology of the navigable zones of the lake. In order to add realism to the global path planning task, several assumptions can be made:The space action of the agent is defined by 8 different actions *a*. Every action, if possible, is equivalent to a waypoint in the adjacent cells. This way, the action space is defined as a∈A:={N,S,E,W,NE,NW,SE,SW}.In both the multi-agent and single-agent case, collision may occur. Three different possible collisions are considered: same-goal collision (two agents move to the same cell), in-way collision (two agents collide in the way to exchange its positions) and an non-navigable collision (the agent intent a movement outside the Lake). In Figure 5, a graphic example of these three possibilities is depicted.The importance map I(x,y) is defined in Figure 3 and discretized depending on the resolution N.One deploy zone is defined for the agent to start (see Figure 3).Every ASV can travel along ∼39.5 km at full speed (2 m/s) considering the battery restrictions. This distance is translated to movements depending on the chosen resolution of the map.It is imposed that the local planner can lead the agent to another cell, different from the desired one, because of control errors and disturbances. There will be always a 5% probability to move to another cell different from the selected one.

## 4. Methodology

This section overviews the two methodologies under study. First, we explain the reward function used to measure the performance of any action. Then, we propose a (μ+λ) Evolutionary method [40] to solve the optimization problem. Finally, we explain the DRL algorithm, the state description, which is treated as a RGB image, and the CNN proposed as a parametric estimator of the action-state Q-function. Furthermore, two multi-agent methodology using in EA and DDQL are also addressed.

### 4.1. Reward Function

In order to evaluate the performance of the methodologies, we have designed a reward function that will serve the purpose of measuring how well the paths adapt to the NHPP. In this particular monitoring problem, the reward function must benefit those actions that lead to a visitation of a high-idle high-importance cell. It must function similarly with the undesirable actions, such as collisions between agents (in the multi-agent particular case) or actions that move ASVs to land cells (unvisitable cells). As the patrolling problem should focus on the useful redundancy of visitations, the weighted idleness W×I of the cell chosen to visit will serve directly as the reward if the action is valid. Thus, visiting a just-visited cell will report a 0 reward as well as visiting a zero-importance one. In order to normalize the positive rewards to a [0,1] interval, the weighted idleness W×I is divided by the maximum possible idleness (maxW) which is, the number of possible movements. This is:(3)$$r_{i}\left(s,a,\left(x,y\right)\right)=\left\{\begin{array}{cc} -5 & \mathrm{if}\;a\;\mathrm{causes}\;a\;\mathrm{move}\;\mathrm{out}\;\mathrm{lake}.\\ -5 & \mathrm{if}\;a\;\mathrm{causes}\;a\;\mathrm{collision}.\\ I_{x,y} & \mathrm{if}\;\left(x,y\right)\;\mathrm{is}\;\mathrm{not}\;\mathrm{discover}\;\mathrm{yet}.\\ \frac{I_{x,y}}{max\;W}\times W_{x,y} & \mathrm{if}\;\left(x,y\right)\;\mathrm{is}\;\mathrm{already}\;\mathrm{visited}. \end{array}\right.$$

In the DRL approach, this reward evaluates every action in every step to optimize the agent’s policy. In the EA approach, the final accumulated value is chosen to be the fitness function, which is:(4)$$R=\sum_{i=0}^{N}r_{i}\left(s_{i},a_{i}\right)$$

The reward function (Equation 3) must reflect how the solution fits the problem to solve, which is, how well each individual of the population fits the design requirements. Whereas the DRL method uses the reward to optimize from one step to the next one and infers the task goal using the sequential reward, the EA only uses the final outcome of the path to evolve its individuals. In the end, paths synthesized by both algorithms can be compared directly using the accumulated episodic reward. In this sense, both algorithms use the reward information differently.

Note that, depending on the resolution, an illegal movement will be more or less criticised. In a lower scale, as an illegal movement implies a higher distance for a useless action, the penalization weights more in the total outcome of the path. In a higher resolution, an illegal movement will be not so terrible since a lower part of the path is lost in this failed attempt. Thus, the reward function defines how desirable, in terms of the addressed problem, an specific action in a specific state for a single-agent is. Then, in this reward function it is specified that the importance of a bad movement is 5 times bigger than the importance of a good action. Attending to the treatment of such bad actions in both methods, we imposed in the DRL that the episode does not terminate when an illegal action is performed so that the agent does not move and the penalization is applied. With the EA approach, it occurs the same and no death penalty over the individuals are applied. This decision is based on two factors: (i) to take advantage of the full episode in DRL, a terminal state limits the exploration of new state-action steps in further steps of this specific episode and (ii) in the EA, the death penalty could discard good individuals because of a single illegal movement in higher resolutions, where illegal actions are more recoverable.

Regarding the design of this reward function, we used the very same function as in [6,25], where a parametrization of the penalization/reward values was proposed. It is remarkable how the design of the reward function affects the stability of the learning. Ref [41] addresses the stability of DRL methods when the range of the reward function is very large. In those previous works, this aspect is analyzed and, here, we used the very same reward function for a fair comparison between EA and DRL, ensuring DRL does not fail because this phenomenon. Actually, this is the very first difference between the DRL and EA: the former has many more parameters to adjust than the latter. EA, as a pure black-box optimizer, does suffer from this kind of problems and this characteristic has to be considered in the following analysis.

### 4.2. Evolutionary Algorithm

The evolutionary algorithms family is a set of meta-heuristic approaches with foundations in the evolution process [42]. These algorithms tend to optimize a population of individuals using an elite selection, crossbreed and mutation operations. Every evolutionary approach begins with a population of random initial individuals. Every population is evaluated with the fitness function and the elite members of this population are chosen to mutate and breed with a probability of MUTPB and CXPB, respectively. In these breed and mutation processes are involved different heuristic operations. In the next generation, the offspring of the previous parents compete with more fitted solutions. This way, iterating over the generations, the population converges to high-fitness paths.

#### 4.2.1. (μ+λ) Algorithm

Following the results of [40], we selected the (μ+λ) EA to optimize the problem as it has been proven to be efficient with a nice convergence for a wide number of problems (see Algorithm 1). In this sense, an individual for the single-ASV case will be represented as a chromosome of T actions (see Figure 6). The first action corresponds to the action taken in t=0 and so on until the last action in t=T. Note that, as the resolution grows, the number of actions increases. The algorithm functioning is as follows:A population of μ individuals is randomly initiated.λ top individuals are selected using a tournament method.The resulting individuals are crossed and mutated with a probability of CXPB and MUTB, respectively, with a two-point crossing (Figure 7) and a K-bit mutation operator (Figure 8).From the extended population formed by (μ+λ), μ individuals are selected by tournament selection to pass to the following generation.

The algorithm iterates through steps 2, 3 and 4 until the upper limit of generations is reached. The stopping criterion is selected depending on the map resolution and to ensure enough convergence. The maximum number of generations is 100, 150, 200 and 250 for the tested resolutions. A higher number of generations has been observed to be inefficient as the solution fitness does not improve.

#### 4.2.2. Multi-Agent EA

Regarding the multi-agent EA approach, a modification is needed to address the case. The individual chromosome is defined now as a set of paths, one for every agent, and the same operators can be applied. Figure 9 shows an individual for a fleet size $F_{s}$ of 3 agents with the actions grouped in temporal order. Consequently, the individual size grows to $F_{s}\times T$, where $F_{s}$ is the fleet size and *T* is the number of possible movements. It is also important to modify the population size as the complexity of the problem scales up. The population will follow the same rule in both, the single-agent and multi-agent case: ten times bigger than the individual size.

### 4.3. Double Deep Q-Learning

Deep Reinforcement Learning is an extension of the well-known Reinforcement Learning algorithms [37]. It uses a combination of the learn-by-trial of RL and Deep Neural Networks to represent the behavior of the agent (in policy iteration methods) or the value function (in value iteration methods). In this manner, by iterating over the domain of actions and states, a estimation of the reward can be made. Deep Q-Learning, a temporal-difference method, is one of the most common approaches of the Temporal-Difference DRL techniques [19] (see Algorithm 2). Those methods are based on predictions of the future reward over successive time steps to drive the learning process. More concretely, DQL uses the Q(s,a) function (action-state function) to estimate the expected reward given a state *s* and an action *a* from the action domain *A*. Given the policy π(s) of the agent, which maps *s* into the probability of taking a certain *a*, $Q^{\pi }\left(s,a\right)$ is formally defined as:(5)$$Q^{\pi }\left(s,a\right)=E\left(\sum_{k=0}^{N}\gamma^{k}r_{t+k}\;|s_{t}=s,a_{t}=a,\pi \left(s\right)\right)$$

The γ term discounts the future behaviors to prioritize the short-term rewards versus the long-term rewards. The objective of Q-Learning is to estimate the optimal value of Q(s,a), the expected reward of every action in the current state, so that the best policy can be followed choosing the action that returns the highest reward value:(6)$$a=\max_{a^{\prime }}Q\left(s,a^{\prime };\theta \right)$$

Due to the dimension of the state domain, the Q-function is usually modeled using a neural network or a convolutional neural network (CNN) if *s* is an image, with parameters θ, instead of a table. This way, the value-iteration method will consists in experiencing several action-state pairs and, by observing the reward *r* and the next state $s^{\prime }$ train the aforementioned network parameters by a stochastic gradient-descent step. To estimate the future expected reward, $Y_{t}^{Q}$ is defined as the target network which is the maximum expected discounted return in the next state. In other words, as DRL is a temporal-difference algorithms, the target function represents an estimated measure of the total amount of reward expected over the future:(7)$$Y_{t}^{Q}=r+\gamma \max_{a^{\prime }\in A}Q\left(s^{\prime },a^{\prime };\theta_{t}\right)$$

Using a stochastic descent step on the gradient direction of a loss function L, the parameters are updated to adjust the Q-function:(8)$$\theta_{t+1}=\theta_{t}+\alpha \frac{\partial }{\partial \theta_{t}}L\left(Y_{t}^{Q},Q\left(s,a\right)\right)$$

This strategy is based on the exploration of many transitions $\left(s,a,r,s^{\prime }\right)$. The behavioral policy must ensure an enough explorative optimization of the state-action domain to learn what is a good action and what is a bad action. The ϵ-greedy policy is an stochastic policy that balances the exploration versus the exploitation of the learned values. It consist on taking a random action with a probability of ϵ and choose a greedy action (the action that reports the higher Q-value) with a probability of 1−ϵ. This way, when a random action is performed, the agent explore allegedly unknown action-states pairs. The value of ϵ is decremented from 1 to a minimum of 0.05 to exploit at the beginning and optimize the estimated gradually. It is common, in order to ensure that learning is uniform across the state domain, to use a buffer replay which stores every experience $<s,a,r,s^{\prime }>$. To learn, a random batch is sampled like in [18], and the network is trained using the gradient descent step method. In this sense, Deep Q-Learning is an off-model algorithm [19], which means it does not need any previous knowledge of the inner dynamics and functioning of the environment to solve and optimize the decision process. This ability will be extremely useful because it can impose several restrictions, boundaries and stochastic dynamics and it will continue without it being necessary to model the system.

Nevertheless, some problems appear when using this methodology. First, as the knowledge is limited to the collected experience, the lack of initial $\left(s,a,r,s^{\prime }\right)$ supposes a slower training in the beginning (cold-start problem). Until the experience replay buffer is full, the network can only learn from a limited number of experiences which can hamper the learning. A possible solution, since the Deep Q-Learning is an off-policy algorithm (the policy that generates the experiences is independent to the behavioral policy of the agent), is to initially full the buffer experience replay with transitions obtained with a completely random policy. For a fair comparison of the sample efficiency of both EA and DRL methodologies, in this work it is not implemented such variations.

The other major problem is that, because the deep neural networks are non-linear it is impossible to guarantee the convergence and stability of the learning. The Deep Q-Learning suffers from several problems like the catastrophic forgetting, which is a sudden deterioration of the performance at some point of the learning process. To improve convergence and learning stability, it was proposed in [19] an improvement called Double Deep Q-Learning (DDQL). In this work, we explained how the max operator in (Equation 7) tends to overestimate the Q-values affecting severely the performance. Thus, a better estimator of the future expected value is chosen to be a temporally displaced function of Q. The target function now uses a copy of Q(s,a;θ) and only updates its parameters once every few episodes. This has been experimentally proven to lead to much better results in the particular case of the Ypacaraí Patrolling Problem [6] and for the ATARI games case [19].

#### 4.3.1. State and Deep Q-Network

DDQL requires a representative state of the scenario in order to infer effectively the causality in the performed actions. Here, the state is represented by an RGB image of the scenario where the different features of the problem can be identified by their colors. The state (see Figure 10) integrates the land cells (brown), the agent position (red) and the visitable cells (gray-scaled). The intensity of the gray visitable cells will depend directly on the weighted idle level W(x,y)×I(x,y) to provide the network with the most available information. Therefore, the temporal dependent events, such as the visitation of a cell, are encoded graphically in the state, so that the first assumption of any MDP is still true. This assumption asserts that the probability to transition from the current state to the next one must only depend on the current state and not on previous events. [37]. Since the state holds past information, the hypothesis is still valid and the learning stability is enhanced.
**Algorithm 1:**(μ+λ)-Evolutionary Algorithm.
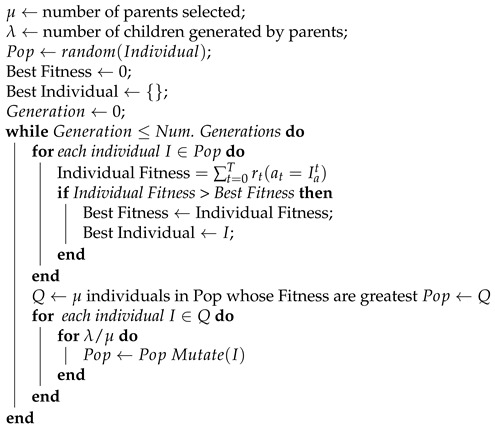

**Algorithm 2:** DDQL with centralized learning.
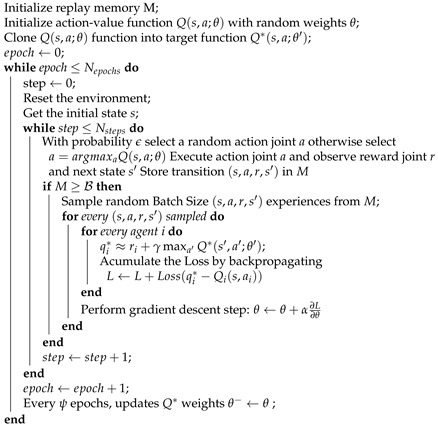


For the Deep Q-function estimator, a CNN is used (see Figure 10) with four dense layers at the end. This architecture was used with satisfactory results in [6,25] in the same Lake problem, so it is selected for this comparison. The network has two parts: (i) a convolutional network at the beginning and (ii) a dense (fully connected) final part. The former analyzes the graphic information and extract the features in the RGB state. The latter receives the features and optimizes the output value, which is the Q-value for every possible action.

#### 4.3.2. Multi-Agent DDQL

The multi-agent Ypacaraí scenario is addressed in [25] with a centralized-learning decoupled-execution DQL algorithm. In this algorithm, every agent has an independent output layer and a shared experience replay (see Figure 11). They share the convolutional dense layers which are optimized with the all the experiences of every agent. In order to share information about the experiences, the state definition changes to include the information of the new agents. Now, the positions of the agents are showed in the RGB state with vivid colors (red, green and blue) to distinguish them from one to another and an extra layer is included to depict only the positions of the agents.

For the optimization step, in the same way as in the single-agent, we computed the loss gradient using the individual reward of each agent using the action in their output layer. This approach, as it uses a common convolutional layer, permits to modify the common learned knowledge using each individual experiences.

## 5. Results and Discussions

In this section, the performance evaluation metrics are explained and a hyper- parametrization of both methods is addressed. The metrics will serve the purpose of measuring how well both algorithms adapt to the requirements. Hence, several experiments have been conducted to test three important problems every monitoring mission faces: different action-state sizes, retraining and reactivity to changes of the scenario. These experiments are explained in the following order: (i) A hyper-parametrization of both methods for the single-agent and lowest-resolution map, (ii) a comparison between the performance of both methodologies in the single-agent case for four different map resolutions, (iii) a retraining and reactivity test in a middle-size resolution with a sudden change of the environment interest map, (iv) the multi-agent performance analysis for a middle-size resolution map, and (v) a generalization test with a changing fleet size.

The different resolution maps that have been tested can be seen in Figure 12. The original continuous map is scaled from the lowest resolution (noted as ×1) to the highest resolution (noted as ×4). These four resolutions correspond with cells of 1 km${}^{2}$ area, 0.5 km${}^{2}$, 0.25 km${}^{2}$ and 0.125 km${}^{2}$, respectively. Consequently, the number of possible steps must be modified. Starting from 30 steps in the ×1 resolution to 60, 90 and 120 in the subsequent maps.

Both algorithms were executed in an AMD Ryzen9 3900 (3.8 GHz) with an Nvidia RTX 2080Super-8GB GPU and 16 GB RAM memory. The simulator was coded in Python 3.8 (https://www.python.org/, accessed on 15 January 2021). For the DRL approach the library PyTorch (https://pytorch.org/, accessed on 15 January 2021) was used and for the EA approach the DEAP (https://deap.readthedocs.io/en/master/, accessed on 15 January 2021) library (Distributed EA in Python) was chosen. The code is available for any purposes on GitHub (https://github.com/derpberk/CEC-RL-versus-EA, accessed on 30 February 2020).

### 5.1. Metrics

To measure the performance of every method, three metrics are used: the Accumulated Episodic Reward (to measure the optimality), the Inverse sample-efficiency rate (to measure the efficiency of the method) and the computation time (to measure the computation requirements).

Accumulated Episodic Reward (AER):The sum of every step reward at the end of a trajectory. When the dimension of the map grows, also does the total available reward.
(9)$$AER=\sum_{t=0}^{T}r_{t}\left(s_{t},a_{t}\right)$$When addressing the multi-agent situation, the AER includes all the individual rewards obtained by the ASVs. Note that, as a higher number of agents causes the cells to be under a higher rate of visitation, the maximum possible reward is not proportional to the fleet size. This happens because the agents share the Lake idleness and the redundancy is inherently higher.
(10)$$AER=\sum_{n=0}^{F_{s}}\left[\sum_{t=0}^{T}r_{t,n}\left(s_{t},a_{t,n}\right)\right]$$Inverse sample-efficiency rate (ISER): This metric consider the number of states observations required for the method to obtain its best solution. It measures the amount of information needed to optimize the paths to a certain optimality level, so, the higher the ISER, the more inefficient is the method because it requires more evaluations of the environment.
(11)$$ISER=\frac{N{}^{\circ}\mathrm{of}\mathrm{Samples}}{AER}$$In the case of the EA, the number of samples is computed as: PopulationSize×Num.ofgenerations×Num.ofsteps. In the case of the DDQL algorithm, it uses the buffer replay and needs extra evaluations of the same states. Considering the algorithm trains the network once every episode with B=BatchSize experiences, the n° of samples is computed as: Batchsize×Num.ofepisodes×Num.ofstepsComputation time (CT): The time that the computer spends optimizing the given solution. On the one hand, for the EA we used 12 cores in a parallel pool to boost the speed. On the other hand, for the DRL approach, a GPU optimized the computation of the gradients. Note that all the available hardware is used to its maximum capacity to minimize the optimization time for the comparison.

### 5.2. Parametrization

For both algorithms it is mandatory to test the performance under the variation of the hyper-parameters that configures the learning and optimization processes [37,43]. As in every machine learning problem and in every optimization application, a systematic approach to tackle the parametrization is not well defined in the literature. As a matter of fact, the results and optimality provided by DRL and EA depends heavily on the selected parameters [44] as it is has been proven in [6,45] or [43], respectively. Even though there are fine-tuning solutions like an extensive grid search, the large number of possible combinations due to the size of the hyper-parameters set scales up, and the time necessary to do such task becomes unfeasible. A too exhaustive search could also result in a very particularization of a certain problem and, when varying the conditions of the task (as it is done in this work), the selected parameters could no longer be valid [46].

Having said that, a preliminary search of a suitable set of values is still mandatory not only to achieve better solutions but also to have an upright comparison. As the number of parameters and possible variations in every method are quite large, especially in DRL where the stability is closely related to them as it was proven in previous works [6,25], only a few of the most important ones are eligible for a realistic parametrization in terms of the searching time. This way, the parametrization has been tested on the lowest resolution (×1 in Figure 12) in a meta-heuristic selection. The use of these hyper-parameters tuned in the lowest resolution are used in further sizes of the map and have been experimentally seen to obtain good solutions even when the state-action domain scales up. This heuristic hyper-parametrization relays on the fact that, since it is impossible to test all parameters (see Table 3) for all the problem casuistic, the lowest resolution must serve as a simplified surrogate model of higher dimension problems.

In the EA, for example, the most sensible parameters have been selected, according to [43]: Crossover Probability (CXPB) and Mutation Probability (MUTPB). These will affect the evolution of the population in the probability to bequeath part of each individual information, and the probability to mutate of every individual to ensure enough diversity, respectively. Other parameters, like μ and λ (chosen to be equal to the population size), are heuristically chosen following hyper-parameter analysis conducted in [43]. The population size also affects the performance: the bigger the population, the more diversity and chances of improvement [42]. The population size has been chosen using a typical rule of the thumb: the population size is ten times bigger than the individual size. For example, with a chromosome size of 30 (resolution ×1), the population size is 300.

Finally, for the following parameter tuning experiment, the values of CXPB are tested from 0.7 to 0.9 and MUTPB is tested from 0.05 to 0.3.

The best convergence falls on the values of CXPB = 0.7 and MUTPB = 0.3, observed in Table 4. A higher value for the mutation probability results in a poorer convergence of the paths and a lower one lacks of exploratory behavior. The same occurs with the crossover probability; with a higher value, the populations tend to fall in local optima.

In the DDQL hyper-parameter tuning, a greedy selection of the parameters has been conducted to solve the highest number of possible combination between these values, like in [6], where the ones that improve the performance over a nominal value are selected. The most sensitive parameters and the selected ones for the tuning are the Learning Rate (LR), the Batch Size (BS) and the ϵ-decay rate. The LR decides the size of the descent-gradient step taken when updating the net’s weights. A big LR could lead the optimization to instabilities and a too low value will cause the net to get trapped in a local optimum. The BS will also affect in the training. The DDQL, as an off-policy algorithm, requires a uniform sample of the previous experiences ($<s,a,r,s^{\prime }>$ tuples). These experiences have to be independent of the policy, which requires a uniform-probability sampling of the Replay Buffer. Higher values of the Batch Size imply training with more experiences, causing the learning process to be more policy-dependent. On the contrary, a low value of the BS causes a slower training [37].

The results, depicted in Figure 13, show that LR and the BS are the most important parameters. With a lower value, the convergence of the DDQL is compromised and a higher value returns also a bad learning process. Thus, the LR and the BS are selected to be $1\times 10^{-5}$ and 64, respectively. The ϵ−decay parameter will balance the exploration of the action-state domain and the explotation of the learned behavior. A low rate implies a more explorative learning and vice versa. The results show that a normalized value of 0.5 (ϵ decays from 1 to 0.05 at the 50% of the total episodes) sets a sufficient exploration for the algorithm. A summary of the final parameters used in the simulations can be found in Table 5 (note that in the EA algorithm, the number of generations changes from a resolution to another because for higher sizes of the map a larger search is needed).

### 5.3. Resolution Comparison

This set of experiments will explore the performance of both methodologies when using different scales. A bigger resolution implies a higher number of state-action combinations and a additional complexity to the problem. This will be important depending on the mission requirements because, in the monitoring task of a water quality variable, a bigger resolution implies the ASV can describe a more spatial-detailed path. This way, the resolution of the problem in different scales is one important test that must be addressed for this NHPP analysis.

In Figure 14, (down) the four experiments conducted for the EA approach are presented. The results show an acceptable convergence of the (λ+μ)-EA algorithm within the given number of generations and it is also represented the results for the DDQL algorithm as well. Notice the neural network converges also in an acceptable policy (see Figure 14 (up)). Besides the fact that performance of both algorithms is similar in the AER metric, these experiments show that the EA approach obtains a sightly superior reward at a fraction of the evaluations (a 2% higher reward) in the simplest resolution. For higher resolutions, the EA approach tends to be computational slower and achieves worse paths. The RL approach obtains a 8%, 4%, and 3% improvement for N = 2,3,4, respectively, much faster (see Table 6). This shows that the EA tends to get trapped in a local-minima and cannot easily achieve better solutions in the considered number of evaluations.

Regarding the average return of the results, the experiments show that, in spite of having a better convergence, DDQL policy returns higher deviations. This is caused mainly because of the high stochastic nature of the ϵ-greedy behavioral policy compared to the elitist approach of the EA algorithm. On average, the standard deviation doubles within the resolution in the DDQL approach whereas in EA remains approximately the same. On the one hand, this suggests that evolutionary approaches can optimize acceptable solutions very similar in fitness from one to another regardless the resolution of the problem. On the other hand, the DDQL approach can explore efficiently every resolution to find better paths but with more sparse solutions. As a matter of fact, this situation of sparser solutions is common in RL approaches (e.g., results in [6,12,18]), where the exploratory behavior is encouraged to provide better convergence but suffers with the augmentation of the action-space dimension.

With respect to the ISER metric and computational time shown in Figure 15, it is noticeable that the sample efficiency of the DDQL is much better than in the EA case. As the complexity of the scenario and the problem dimension grows, it is necessary to augment the number of generations and individuals to achieve acceptable results in the episodic reward. For its part, the DDQL can generalize across the resolution with few evaluations. With an ISER to 37.5% and 71% lower for N = ×3 and N = ×4, respectively, DDQL results in a very sample efficient methodology to optimize very complex problems. The time consumption will depend directly on the hardware. As it has been said before, the use of GPUs allows the DDQL to speed up the execution of the optimization compared to the parallelization execution of EAs. Nonetheless, this metric is only representative of the algorithm performance when measured using components (CPU and GPU) similar in cost, like the ones used here.

### 5.4. Scenario Reactivity

For this particular problem, it is interesting to compare the ability of the algorithms to react to a changing scenario. As it is mentioned before, the interest map could experience changes due to the dynamic nature of the blue-green algae. In this way, both algorithms can be compared by its aptitude to adapt the optimization to a different fitness function. This is an usual situation in the monitoring requirements, because between missions or in the middle of one, the interest map could variate significantly and a retraining will be mandatory.

In the following experiment, both algorithms must face a different interest map (with resolution N = ×2 for simplicity) after they optimize the previous scenario (see Figure 16 for a representation of the new scenario). This situation simulates a casuistic where the previous knowledge (represented by experiences and individuals) of the scenario must be used to synthesize a new solution. In the simulation it is specified that, after 1500 episodes or 150 generations (cases with the same sample-efficiency; see Section 5.1), the scenario changes and a re-optimization is needed. At this point, two different variations are tested for every methodology: the EA can maintain its previous population, taking advantage of previously experienced paths, or reinitialize randomly a new population. Similarly, the DDQL approach, can empty the Buffer Replay of past experiences, or continue learning from past experiences no matter that they do not corresponds with the actual scenario.

The results (see Figure 17) are different depending on the approach and variation. In the DRL case, it can be observed how the adaptability and retraining process is slightly better without emptying the buffer. Both variants converge in the same solution with a maximum reward of 45.4, but emptying the buffer only causes a delay in the learning process. As the new scenario does not variate excessively from the previous, the old experiences $<s,a,r,s^{\prime }>$ are still valid for training and including them speed up the learning process. In the EA counterpart, a maximum reward of 33.5 is reached when using the previous population. The generalization is excessively complicated to optimize the paths because of the lack of diversity in its previous population. It is mandatory to regenerate the population in order to have good convergence. When this is done, a maximum reward of 44.3 is obtained. As a summary, DRL obtains a 2% better solution in the second scenario and shows a better re-learning capabilities once the previous task is generalized.

### 5.5. Multi-Agent Comparison

The multi-agent case is a generalization of the single-agent case applied to a fleet size different from one. This case addresses the situation of multiple ASVs deployed in different starting points (see Figure 18). As Ypacaraí Lake is big, the use of multiple vehicles helps to cover more quickly the water. Nonetheless, the coordination task must be addressed by the optimization algorithm apart from the difficulties of resolving the patrolling problem. Using the methodology explained in Section 4, both algorithms will attempt the optimization with different fleet sizes $F_{s}$ from 1 single agent (as in previous subsection) to 3 agents. The objective of this set of experiments is to analyze how both methodologies reacts to an augmentation of the action domain.

For the EA approach, the population size must be modified consequently to the new action domain as it is explained in Section 4: with the increase of $F_{s}$, the population size of the EA is increased so that it is ten times bigger than the individual size. The population size will be 1200 and 1800 for $F_{s}=2$ and $F_{s}=3$, respectively. In the DRL approach, the hyper-parameters remain the same as the proposed network is designed so that the execution and learning of the agents occur independently. Regarding the number of epochs and generations evaluated, as it is intended to compare the performance with the single-agent scenario as a baseline, the same number of epochs/generation from the single-agent case is imposed. In Table 7, the parameters used in this three simulations are summarized.

The results in Figure 19 show that the performance of DDQL depending on the fleet size is higher than the obtained optimizing with the EA. For the multi-agent case of $F_{s}=2$, the performance is improved by 27.1 % which indicates a much better performance within the size of the fleet. In the largest fleet size $F_{s}=3$, the DDQL also improves the EA paths by 21.5% compared to the maximum reward. This clearly indicates the dimension of the individual significantly affects the performance of the evolutionary approach. On the contrary, as it happened in the previous experiment, the deviation of the DDQL grows within the dimension in a higher rate than in the EA (see Table 8 for all the statistics). This is a common phenomena in the DRL policies because of the stochastic learning process [37] of the behavioral function. This situation is more severe when the multi-agent case is addressed because the RL faces not only the unstable learning challenge but also the credit assignment problem and the non-stationary of the environment [47]. This triad of problems demands a higher number of experiences to synthesize a more robust policy in spite of the good maximum performance achieved for the best trajectory.

With respect to the sample efficiency, in Figure 20 it can be seen that the DDQL approach is a better solution for higher action spaces. It is remarkable how the DDQL algorithm is able to obtain good solutions with the very same amount episodes whereas the EA easily struggles with local-optima solutions. In order to obtain these sub-optimal solutions, it was mandatory to impose a bigger size of the population to improve the diversity and exploration but, in spite of the many evaluations, the black box approach used in EA cannot outperform DDQL. The computation times is also higher in the EA. The GPU computation is able to resolve and optimize the convolutional neural network an order of magnitude faster than the CPU calculation used in the EA.

### 5.6. Fleet Size Generalization

In a practical implementation, it is always a possibility that one or more ASVs are unable to accomplish the mission. This could be because of a battery fail, the initial deploy zone is unavailable, etc. Therefore, when this occurs, the pre-calculated paths are no longer optimal. The algorithms must be compared by its ability to propose acceptable solutions in this particular situation. This way, in the last experiment, we intended to compare the generalization abilities of both methods when the fleet size suddenly changes. For this experiment, paths for a fleet size of 2 ASVs are optimized starting from positions 1 and 2 like in Figure 18. Once the optimization ends, the best paths of EA is compared in performance to the those generated in DDQL.

Note that in the DDQL, the state will no longer contain the ASV n° 2 once it is deactivated. The deep policy is expected to change the behavior of the agent n° 1 since it notices that there is no other agent to cooperate with. The EA paths, on the contrary, are optimized without any reactive nor cooperative criteria and will stay the same independently of the changes in the scenario and its performance relays only on each agents own path. To compare the results between algorithms and between the single-agent case, the $R_{max}$ for $F_{s}=1$ is used as a baseline to measure how much the MA-trained policies adapt to the single-agent paradigm.

As expected, the results in Table 9 sustain the predicted reactive behavior of EA and DDQL. The former case achieves a maximum fitness $R_{max}^{F_{s}=2\to 1}$ of 36.4 which is 83.5% of the fitness in the single-agent case (recovery rate). The DDQL obtains a maximum reward of 41.92 which is a recovery rate of 90% compared to the single-agent scenario record. This indicates that the task is well generalized by the DRL approach, as the other agent’s disappearing causes the deep policy to the uncovered zones. It is important to highlight that in the DDQL approach, there is not a full generalization of the task, since the result indicates there is room for improvement. This is a natural consequence of every DRL process, since unexpected states may lead to sub-optimal decisions because of the lack of previous experiences. As a matter of fact, the EA proposes non-reactive but robust solutions whereas, with DRL approaches, there is always the risk of instabilities and a higher level of uncertainties.

### 5.7. Discussion of the Results

It has been proved that DDQL and EA can solve the Non-Homogeneous Patrolling Problem. DDQL is a very efficient algorithm for solving high dimensional problems such as the Ypacaraí path planning task, nonetheless, EA is still better in some aspects. After a comparison between the two methodologies some considerations can be made regarding the action-state dimension, the reactivity to scenario changes and the multi-agent case. Finally, in Table 10 it is included a summary of the advantages and drawbacks of both algorithms.

Attending to the resolution of the map, both algorithms return acceptable results. It is noticeable that both algorithms obtain very similar maximum rewards. Nevertheless, the efficiency of the optimization changes depending on the size of the map. The results show that, for higher resolutions, EA algorithms tend to be slower and inefficient as the population size needs to be large and several generations are needed. DDQL is proven to perform great in every tested map, which indicates that is a very suitable methodology in high dimension problems. Hence, Figure 15 could serve to choose the methodology depending on the size of the environment in similar problems.As for the multi-agent case, it can be noticed how the number of agents affects more significantly than the map size. The DDQL outperforms the EA in approximately a 25%, which implies the former is a better algorithm to deal with multiple agents. As a matter of fact, the black box methodology of the EA cannot take any advantages from the scenario via the state, whilst the DDQL can effectively infer some amount of knowledge from every action taken.Comparing the robustness of the proposed solutions, the EA, in spite of returning worse paths, can provide a set of solutions with a much lower deviation. DDQL, as it uses a stochastic approach, tends to generate very bad paths sometimes in an example of the catastrophic forgetting phenomenon. The dispersion of the results aggravates when the shape of the map or the fleet size grows and more experiences are needed for a better policy convergence. Nonetheless, when the scenario changes, either because the interest map changes or the size of the fleet variates, the reactivity of the DDQL is able to adapt the policy better. This is because the DDQL method, as it learns from the data and not only the final outcome, learns to perform the task.not only the reward function, which implies a similar mission can be completed effectively.It is important to note that the convergence of the DDQL is not guaranteed in any case because of the use of non-linear estimators as Neural Networks [19]. Therefore, the DDQL depends on the suitable hyper-parameter selection, network architecture, etc, to optimize correctly the solutions. This is an important aspect to consider because, in spite of the good results, EA does not need a critic design to provide similar results in most cases.

## 6. Conclusions and Future Work

In this work, two methodologies such as EA and DDQL have been used to solve the same problem: the Ypacaraí Patrolling Planning. As an NP-hard problem, both methods are selected due to their known abilities to deal with high dimensions. It has been observed that DDQL and the classic (μ+λ)-EA can effectively resolve the NHPP within many important dimensional variations of every path planning task: the size of the environment, the size of the fleet or sudden changes in the problem to solve.

Furthermore, this study has analyzed how the size of the map affects the performance in both methodologies, resulting in a better and faster optimization for the Evolutionary Approach for lower dimensions. For environments with much higher number of states, the EA approach tends to be slower and inefficient, but with a similar convergence. The DDQL, because its ability to learn information from the collected data (and not only from the direct outcome of the fitness function), excels in this high dimensionality problems. This happens also when, given a specific map size, the fitness function changes because of a dynamic interest criterion. The DDQL showed a nice capacity to learn the intended task and, consequently, its ability to perform in a different scenario outperforms the results of the EA.

We can also conclude that regarding the stability and performance, EA is a very robust approach with respect to the hyper-parameters selected. DDQL (and other DRL approaches) must face learning stability issues and depends heavily on the hyper-parameters selected for this particular application [6]. In this sense, it will be interesting, in terms of a more efficient hyper-parameter tuning, to explore the use of Bayesian Optimization techniques [48] which can provide an acceptable set of hyper-parameters with less tuning experiments. These meta-learning methodologies are promising and cannot be neglected in a real implementation scenario, where the task are performance-critic.

It is the designer’s criterion to choose the most suitable approach, considering not only the scale but also the capacity of the given method to optimize well. Depending on the application, the on-line recalculation of the whole problem could be impossible having the computation times above and a sub-optimal solution could be the only possibility. In this situation, DDQL can return better behaviors but it is impossible to prove a guaranteed convergence due to the intrinsic DRL mathematical functioning. Thus, EAs may serve as off-the-shelf path planning algorithms ready to optimize any problem no matter the specifications if the time requirement is not strict. These methodologies can also be used to solve the monitoring problem for different lakes as the nature of the algorithms is intrinsically off-model. It is advisable, in order to obtain a suitable solution in a decent amount of time, to scale the map in a resolution no much longer than the studied in this work and to perform a hyper-parameter study in any case. For the interest map, a relatively soft function of interest peaks has been used. This softness is encouraged because it provides a more continuous reward function, easily to assimilate by the DRL, and can suit easily the environmental monitoring criteria in other cases apart from Ypacaraí.

Nonetheless, in spite of being EA and DRL independent of the environment, a further analysis on the stability and convergence of this kind of problems is necessary, especially when addressing different tasks in other similar scenarios. Future works will put the attention in a learning stability study and how the data, the reward function and the constraints of the problem affect the final performance. When addressing the computational efficiency and execution-time reduction, it will be interesting to explore the possibility of clustering the different zones of interest in hierarchical missions. Recent approaches like [49] suggest this modification of the problem, equally applicable to EA and DRL, provides with much lower computation times. In this sense, a wider study on how the performance of both optimization techniques are affected by the hardware is needed. The use of GPU has been proved to boost up the DRL time-efficiency and it is mandatory at this point to explore the possibility to use similar hardware-enhanced optimization for the EA counterpart.

## Figures and Tables

**Figure 1 sensors-21-02862-f001:**
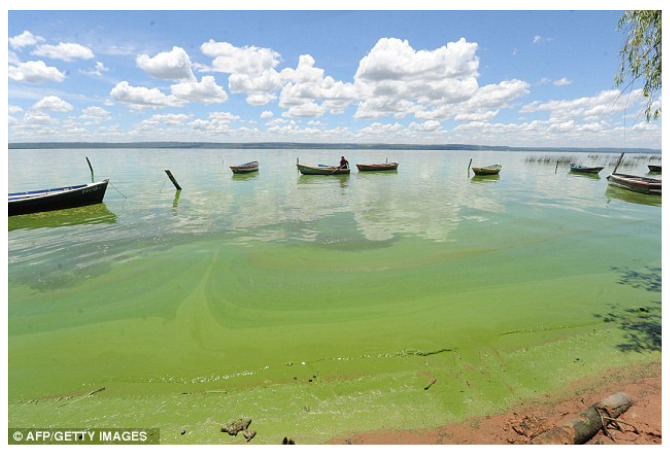
Cyanobacteria effects on the Lake shore. The contaminated water has a characteristic intense green color and gives off an unpleasant smell.

**Figure 2 sensors-21-02862-f002:**
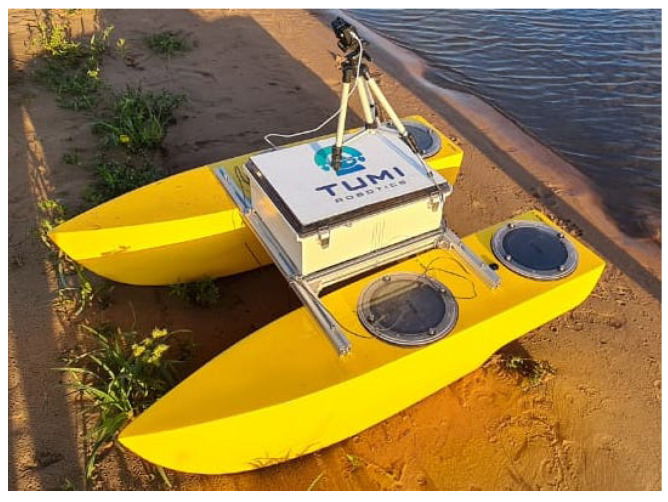
Autonomous Vehicles prototype used in [3] for the monitoring of Ypacaraí Lake. The vehicle has two separate motors and a battery that allows a stable speed of 2 m/s during 4 h of travel.

**Figure 3 sensors-21-02862-f003:**
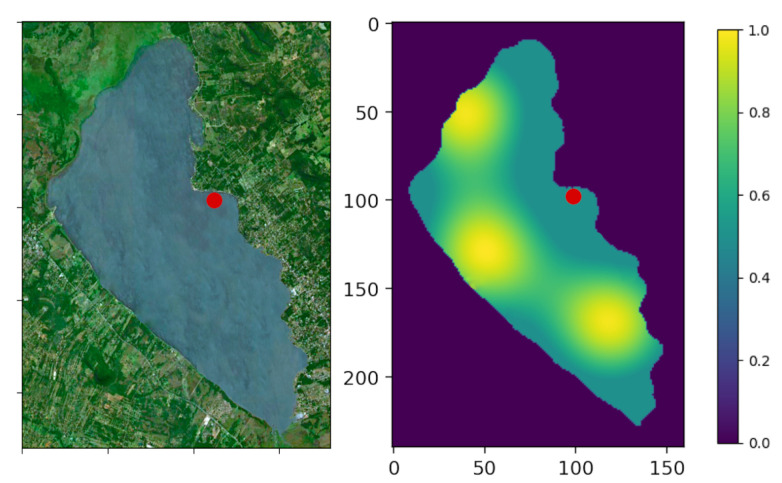
The left image shows a satellite image of Ypacaraí Lake. In red, the initial deploy zone for the ASV. The right image shows the importance map I(x,y) that weights the idle index of every zone. The higher the value of I(x,y) the most important is to cover this area, either because it is a very contaminated area or holds a high biological interest.

**Figure 4 sensors-21-02862-f004:**
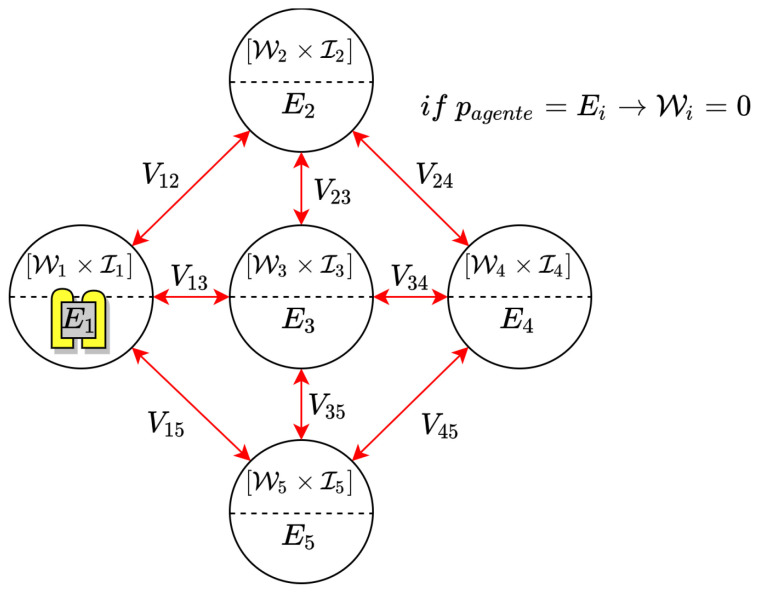
Example of graph G(E,V,W) for the Patrolling Problem. There is a metric assumption and every cell will hold its idleness *W* as the number of time steps since the last visit. In the beginning, the value of *W* in every cell is $W_{max}$.

**Figure 5 sensors-21-02862-f005:**
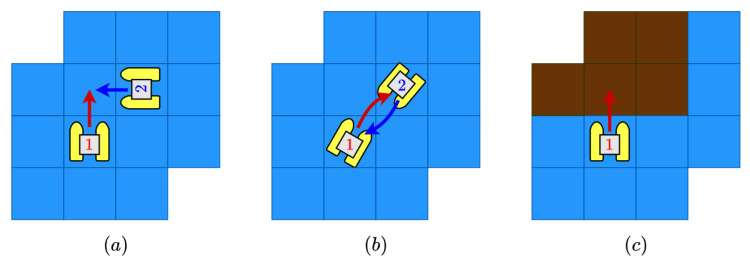
Movements considered collisions in the multi-agent scenario. Every collision is rewarded with a negative penalization. If the collision involves more than one agent, each agent is independently penalized. (**a**) A same-place collision is depicted. (**b**) A in-transit collision. (**c**) A off-water illegal movement can be seen.

**Figure 6 sensors-21-02862-f006:**
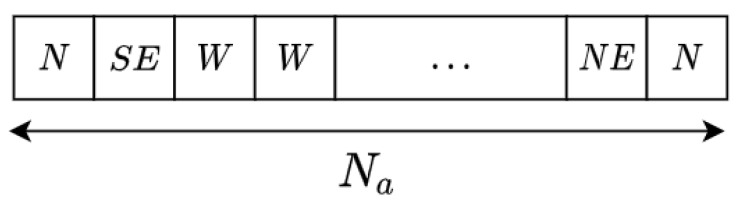
Individual representation. Every individual is a list of *T* actions. One for every timestep *t* of the trajectory.

**Figure 7 sensors-21-02862-f007:**
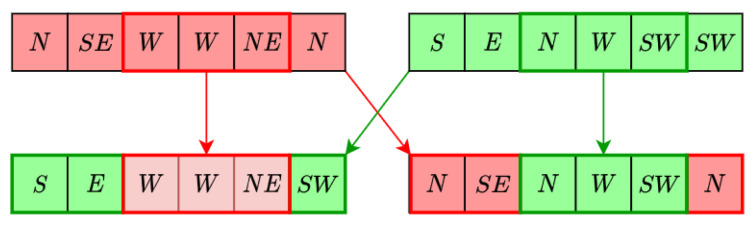
Two point crossing operation of two individuals.

**Figure 8 sensors-21-02862-f008:**
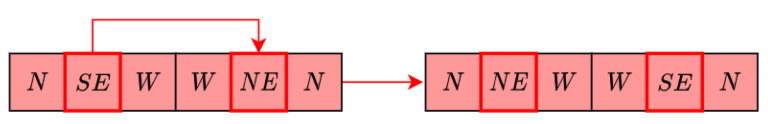
1-bit mutation of an individual.

**Figure 9 sensors-21-02862-f009:**
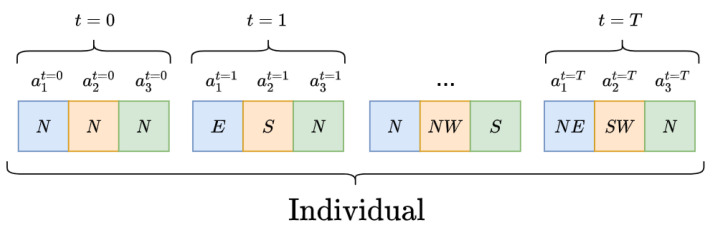
The multi-agent individual representation for $F_{s}=3$ agents. Actions are grouped three by three for every timestep t∈[0,T].

**Figure 10 sensors-21-02862-f010:**
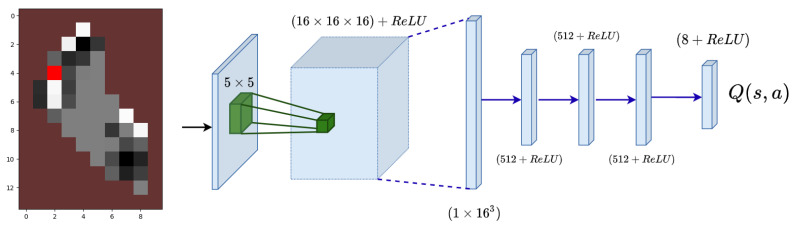
In the left, the state of the environment. In red, the agent position. In brown the land cells. In gray scale, the visitable zones. In the right, the Convolutional Neural Network. The convolutional layers converge in dense layers using the Re-LU function as an activation layer for every neuron.

**Figure 11 sensors-21-02862-f011:**
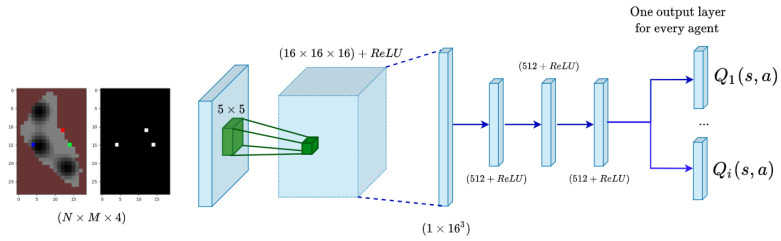
Centralized-learning decoupled-execution network for the multi-agent approach in [25]. Note that the state (left) indicates with vivid colors the different agents positions and an extra channel is added for a better positioning in the collision avoidance task.

**Figure 12 sensors-21-02862-f012:**
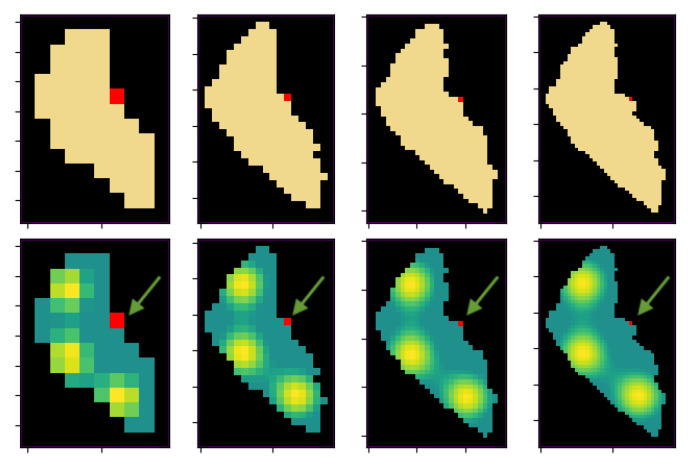
Different discretized maps of Ypacaraí Lake with N = ×1, ×2, ×3 and ×4 resolution. In red, the initial point of the agent.

**Figure 13 sensors-21-02862-f013:**
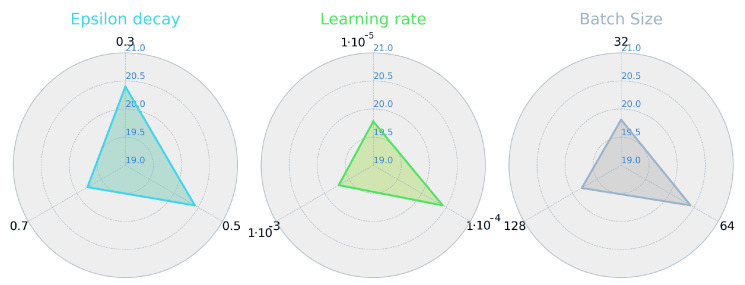
Max. reward obtained for the greedy parametrization in the DDQL approach.

**Figure 14 sensors-21-02862-f014:**
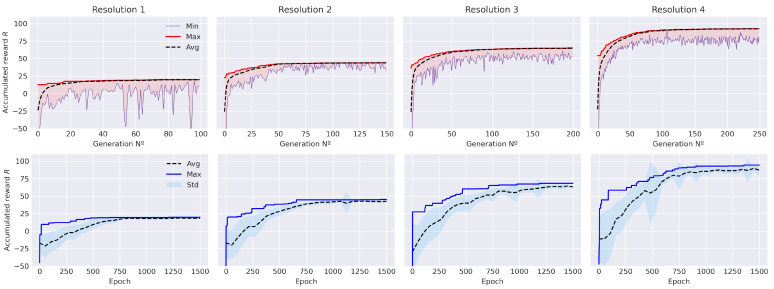
Accumulated reward along the training process in the EA approach (**up**) and in the DDQL approach (**down**). Note in the EA, every data corresponds to a generation whereas in the DDQL corresponds to an episode.

**Figure 15 sensors-21-02862-f015:**
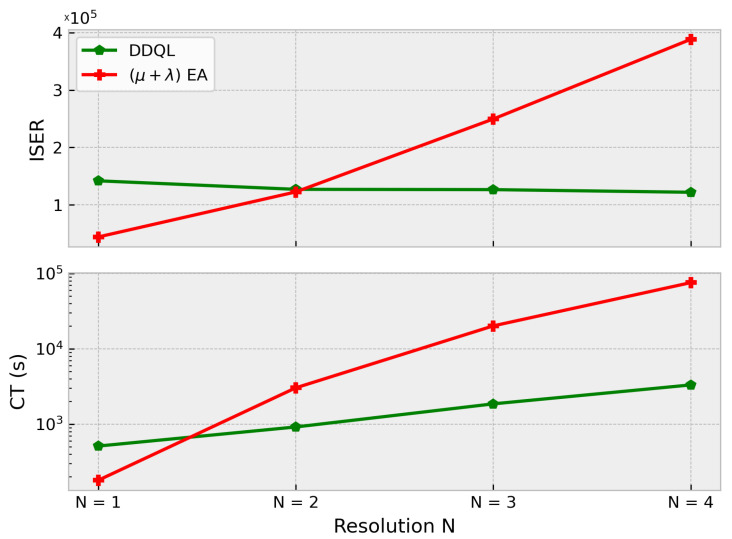
ISER (**up**) and CT (**down**) for each approach and every resolution N.

**Figure 16 sensors-21-02862-f016:**
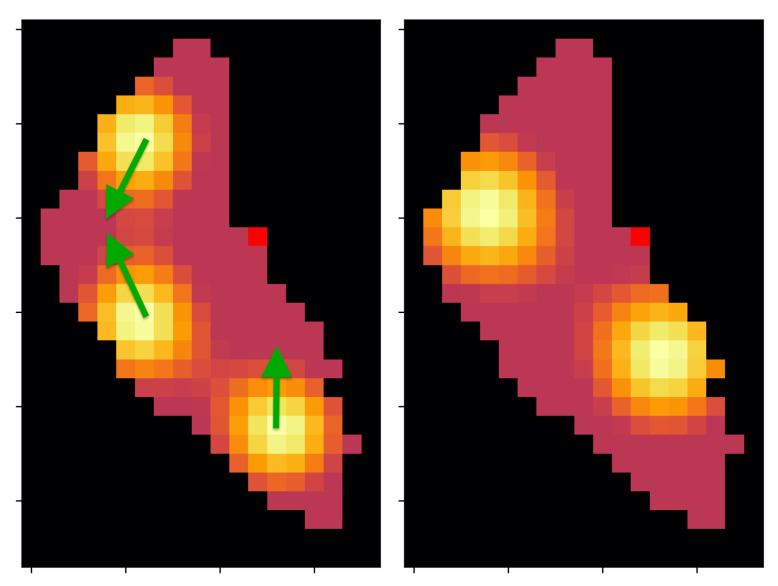
Transformation of the interest map. The green arrows represents the movements of the peaks of interest.

**Figure 17 sensors-21-02862-f017:**
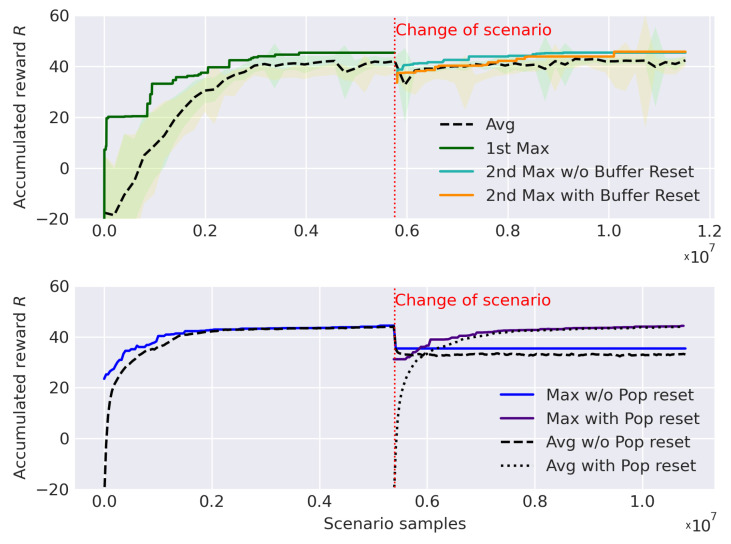
Result of the optimization with a change in the interest map for DDQL (**up**) and EA (**down**).

**Figure 18 sensors-21-02862-f018:**
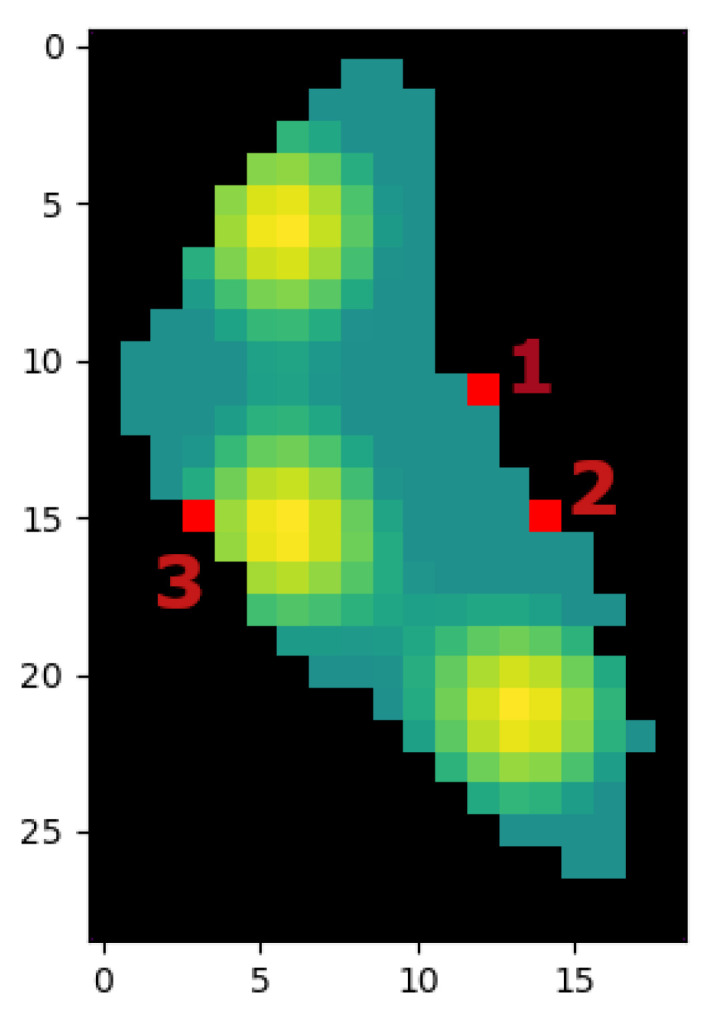
Different initial points (red) in the N = ×2 resolution map. Each ASV is assigned with one specific initial point, so the order in the EA chromosomes and each output layer of DRL will be the same from a generation/episode to another.

**Figure 19 sensors-21-02862-f019:**
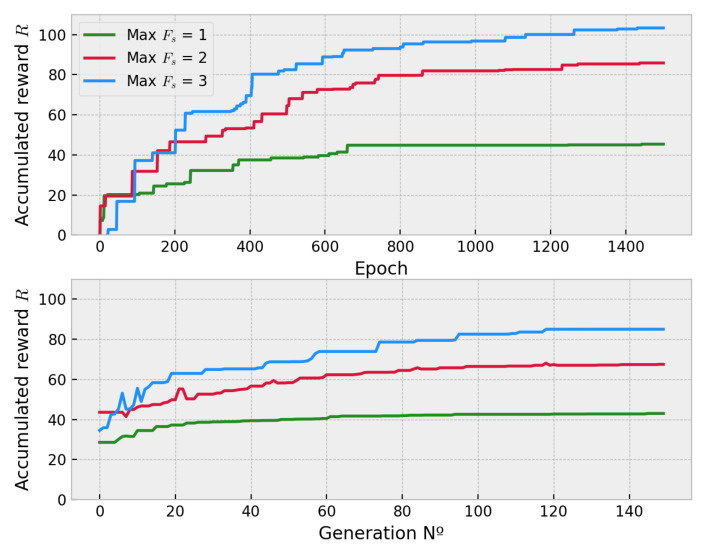
Training progression of the max. AER in the multi-agent case for DDQL (**up**) and EA (**down**) for 1500 episodes and a max. value of generations of 150 generations.

**Figure 20 sensors-21-02862-f020:**
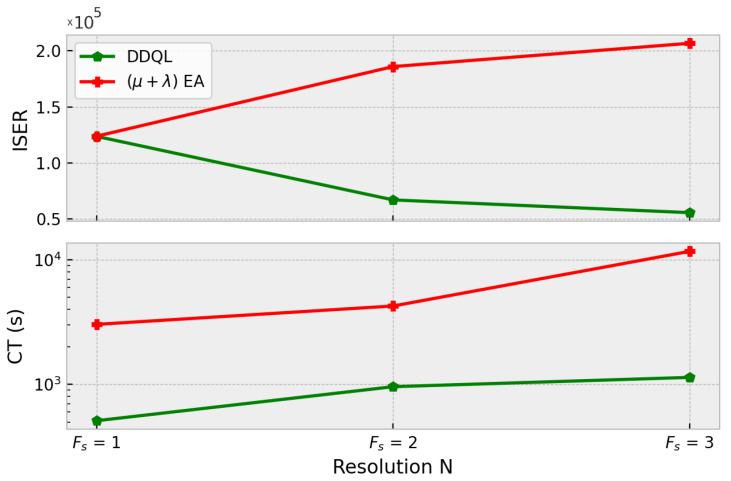
ISER (**up**) and computation time (**down**) for different fleet sizes. When the number of agents increases, the EA cannot find good solutions with the same amount of samples as the DDQL.

**Table 1 sensors-21-02862-t001:** Brief summary of path planning using DRL.

Ref.	Objective	Methodology	Vehicle
[12]	Informative Path Planning.	Deep Q-Learning with CNNs.	single-agent UAV
[20]	Complete Surface Coverage in cleaning tasks.	DRL Actors Critic with Experience Replay	multi-body single ASV
[21]	Coverage Maximization with FANETs.	Deep Q-Learning.	single-agent UAV
[22]	Docking Control and Planning.	Deep Q-Learning and DDPG	single-agent ASV
[6]	Patrolling Problem.	Deep Q-Learning with CNNs	single-agent ASV
[25]	Multi-agent Patrolling Problem.	Deep Q-Learning with CNNs	multi-agent fleet of ASVs
[27]	Multi-agent Path Planning with obstacles avoidance.	Deep Q-Learning with CNNs	multi-agent fleet of UAVs
[28]	Multi-agent Path Planning with obstacles avoidance.	Deep Q-Learning with CNNs	multi-agent fleet of UAVs
[29]	Multi-agent Urban Path Planning.	Proximal Policy Optimization.	multi-agent fleet of unmanned cars

**Table 2 sensors-21-02862-t002:** Brief summary of path planning using EAs.

Ref.	Objective	Methodology	Vehicle
[2]	Coverage TSP global Path Planning.	Evolutionary Algorithm.	single-agent ASV
[7]	Homogeneous Patrolling Problem with grid-maps.	Evolutionary Algorithm.	single-agent and multi-agent robots.
[31]	3D optimization of an UAV paths.	Evolutionary Algorithm.	6-DOF single-agent UAV.
[15]	Multi-joint path-planning in manipulator robots.	Evolutionary Algorithm	single-agent robot.
[32]	Multi-agent cooperation in path planning.	Natural-Language Evolutionary Strategy.	multi-agent fleet of robots.
[33]	Multi-agent in path planning with dynamic scenario.	Evolutionary Algorithm.	multi-agent fleet of robots.
[34]	Multi-agent cooperation-competition in path planning.	Evolutionary Algorithm.	multi-agent fleet of robots.
[35]	Multi-agent goal-tracking in path planning.	Novel Genetic Algorithm.	multi-agent fleet of UAVs.

**Table 3 sensors-21-02862-t003:** Main hyper-parameters involved in both methods. With *, the most sensible hyper-parameters for each algorithm.

Hyper-Parameters	
***DDQL***	**(μ+λ)-EA**
- γ - Batch Size * - ϵ-decay rate * - $\epsilon_{min}$ - Learning Rate (α) * - Target update rate (ψ) - Epochs - Buffer Replay Size (M) - Network Arquitecture	- MUTPB * - CXPB * - Tournament Size - μ+λ - Population Size - Generations

**Table 4 sensors-21-02862-t004:** Maximum reward result of the parametrization in the EA. In bold, the best values between both algorithms.

	CXPB	0.9	0.8	0.7
MUTPB				
0.05		18.02	18.41	18.66
0.1		18.48	19.69	18.85
0.2		-	19.98	19.96
0.3		-	-	**20.4**

The bold indicates the best result between both methods.

**Table 5 sensors-21-02862-t005:** Parameters for both approaches.

EA		DDQL	
(μ+λ)	(300, 600, 900, 1200)	γ	0.95
MUTPB	0.3	ϵ-decay	$6.5\times 10^{-3}$ (0.5)
CXPB	0.7	Batch Size	64
Generations	(100, 150, 200, 250)	Episodes	1500
Population Size	(300, 600, 900, 1200)	Learning rate	$1\times 10^{-4}$

**Table 6 sensors-21-02862-t006:** Experiments results for EA and DDQL algorithms when optimizing with four different resolutions of the map. In bold, the best results of both methods.

	(λ+μ)-EA			DDQL		
	$R_{max}$	$\bar{R}$	$\sigma_{R}$	$R_{max}$	$\bar{R}$	$\sigma_{R}$
N = ×1	**20.4**	19.8	0.78	20.3	17.7	0.99
N = ×2	43.2	42.7	0.56	**46.6**	43.9	1.81
N = ×3	65.9	65.3	0.41	**68.3**	64.5	2.35
N = ×4	93.6	92.2	0.63	**94.7**	89.7	3.78

The bold indicates the best result between both methods.

**Table 7 sensors-21-02862-t007:** Parameters for the multi-agent case simulations.

EA		DDQL	
μ,λ	(600, 1200, 1800)	γ	0.99
MUTPB	0.3	ϵ-decay	$6.5\times 10^{-3}$ (0.5)
CXPB	0.7	Batch Size	64
Generations	150	Episodes	1500
Population Size	(600, 1200, 1800)	Learning rate	$1\times 10^{-4}$

**Table 8 sensors-21-02862-t008:** Experiments results for multi-agent case in EA and DDQL algorithms when optimizing with N = ×2 for 150 generations in EA and 1500 episodes in DDQL.

	(λ+μ)-EA			DDQL		
	$R_{max}$	$\bar{R}$	$\sigma_{R}$	$R_{max}$	$\bar{R}$	$\sigma_{R}$
$F_{s}$ = 1	43.6	42.7	0.56	**46.6**	43.9	1.81
$F_{s}$ = 2	67.5	62.1	9.3	**85.85**	73.3	27.09
$F_{s}$ = 3	85.01	75.1	27.7	**103.32**	75.2	32.26

The bold indicates the best result between both methods.

**Table 9 sensors-21-02862-t009:** Experiments results for multi-agent case in EA and DDQL algorithms when optimizing with N = ×2 and the agent starting from position 2 is not available for the mission.

	(μ+λ)-EA	DDQL
$R_{max}^{F_{s}=2\to 1}$	36.4	41.9
$R_{max}^{F_{s}=1}$	43.6	46.6
Recovery	83.5%	90%

**Table 10 sensors-21-02862-t010:** Summary of the characteristics of DDQL and EA applied to the NHPP.

	*DDQL*	(μ+λ)-EA
**Performance**	Very high convergence in every resolution.	Better convergence in low resolutions. Suffers from dimensionality with higher scales.
**Robustness**	As a stochastic approach, suffers from robustness. The solutions shows a much higher dispersion.	The population has a lower dispersion in the optimality.
**Sample-efficiency**	Very sample efficient. It can synthesize excellent solutions with very few samples.	Need more samples and evaluations of the environment.
**Computation time**	Using GPUs, excels in the computation time efficiency.	Using CPU parallelism, is faster for lower resolutions. For higher resolutions, the computation time becomes unfeasible.
**Reactivity**	Excellent reactivity to changes in the environment. Can retrain faster when the dynamics or the state changes. May lead to unexpected outputs due to a biased model.	Non-reactive at all.
**Parametrization**	Several hyper-parameters to tune. The neural network must be tested profoundly. Some parameters could lead to instabilities of the learning process. The reward function must be designed appropriately.	Not many important parameters. Very robust to the parameters. They can affect the performance but not prone to compromise the stability. Can optimize regardless the reward-function.

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
