# Peer review of "A Dimensional Comparison between Evolutionary Algorithm and Deep Reinforcement Learning Methodologies for Autonomous Surface Vehicles with Water Quality Sensors"

_sensors, 2021, doi:10.3390/s21082862_

Round 1

Reviewer 1 Report

The proposed manuscript is very interesting because it compares two common planning technologies (Evolutionary Algorithms and Deep Reinforcement Learning) with different parameters that may affect performances and results.

The paper is well written and presented, except for very minor flaws. It is easy to read and easy to follow. To me, very few concerns are to be pointed out to authors:

  1. In the abstract the rows 3-4 to me are not clear: the contamination problem of the Ypacarai Lake is due to the onboard sensor modules?
  2. In algorithm 1 the first while cycle evaluates "e<=epochs". What's "e"? Is not included in the algorithm
  3. Row 505: "USED" is written in capital letters. Is that right?
  4. As cited by the authors, in [28] the same problem has been covered with a non-stochastic method, the Traveling Salesman Problem. Due to the high computational times of EA and DDQL (more than a half a minute - at least - as reported by the authors), it should have been interesting to include TSP as an additional methodology (since it is also possible to drastically reduce its computational time by clustering the possible target positions, see [a] below). Please consider this method for future applications (maybe in the future work section), since with little computational times it is possible to face interest map changes in the middle of missions, a scenario which is possible as stated in rows 634-636.

[a] Bottin M., Rosati G., Boschetti G. (2021) Working Cycle Sequence Optimization for Industrial Robots. In: Niola V., Gasparetto A. (eds) Advances in Italian Mechanism Science. IFToMM ITALY 2020. Mechanisms and Machine Science, vol 91. Springer, Cham. https://doi.org/10.1007/978-3-030-55807-9_26

Author Response

Thank you.

Reviewer 2 Report

This paper proposed a supervised RL strategy for autonomous driving via virtual safety cages. The paper is generally well written with sufficient analysis. It can be reconsidered for acceptance if the authors can address the following issues.

  1. RL-based decision has been widely used for vehicles, e.g., DOI: 10.1109/TII.2020.3014599; DOI: 10.1109/TVT.2020.3025627; Journal of Power Sources 489 (2021): 229462. These works can be mentioned to improve the literature review and method description.
  2. The RL-based policy may be subject to the cold start-up problem, i.e., the early-stage learning will be hard due to the exploration of unfamiliar environment. This should be commented in the paper.
  3. Both the EA and the RL depend largely on the hyperparameter setting. So please provide necessary information on this.

Author Response

Thank you.
